# Genetic variant in 3' untranslated region of the mouse *pycard* gene regulates inflammasome activity

**Brian Ritchey[1†], Qimin Hai[1†], Juying Han[1], John Barnard[2], Jonathan D Smith[1,3]***

[1]Department of Cardiovascular & Metabolic Sciences, Lerner Research Institute, Cleveland Clinic, Cleveland, United States; [2]Department of Quantitative Health Sciences, Lerner Research Institute, Cleveland Clinic, Cleveland, United States; [3]Department of Molecular Medicine, Cleveland Clinic Lerner College of Medicine of Case Western Reserve University, Cleveland, United States

**Abstract** Quantitative trait locus mapping for interleukin-1β release after inflammasome priming and activation was performed on bone-marrow-derived macrophages (BMDM) from an AKRxDBA/2 mouse strain intercross. The strongest associated locus mapped very close to the *Pycard* gene on chromosome 7, which codes for the inflammasome adaptor protein apoptosis-associated speck-like protein containing a CARD (ASC). The DBA/2 and AKR *Pycard* genes only differ at a single-nucleotide polymorphism (SNP) in their 3' untranslated region (UTR). DBA/2 vs. AKR BMDM had increased levels of *Pycard* mRNA expression and ASC protein, and increased inflammasome speck formation, which was associated with increased *Pycard* mRNA stability without an increased transcription rate. CRISPR/Cas9 gene editing was performed on DBA/2 embryonic stem cells to change the *Pycard* 3'UTR SNP from the DBA/2 to the AKR allele. This single base change significantly reduced *Pycard* expression and inflammasome activity after cells were differentiated into macrophages due to reduced *Pycard* mRNA stability.

**\*For correspondence:** smithj4@ccf.org

[†]These authors contributed equally to this work

**Competing interests:** The authors declare that no competing interests exist.

## Introduction

Genetic differences between inbred mouse strains have facilitated the discovery of many disease genes and pathways, as well as modifier genes, through the use of quantitative trait locus (QTL) mapping and molecular technologies. Our group has focused on the AKR and DBA/2 mouse strains to interrogate genes and pathways associated with atherosclerosis, along with several macrophage phenotypes that may contribute to atherosclerosis pathology. We previously observed that 16-week-old chow diet-fed DBA/2 vs. AKR mice develop ~10 fold larger aortic root atherosclerotic lesions after breeding hyperlipidemic apoE knockout mice onto these distinct genetic backgrounds. This divergent atherosclerosis phenotype was observed in both males and females (*Smith et al., 2006*). Two independent strain intercrosses were performed and QTL analysis identified three significant loci associated with lesion area, called *Ath28*, *Ath22*, and *Ath26*, on chromosomes 2, 15, and 17, respectively (*Hsu and Smith, 2013*). We subsequently identified the *Cyp4f13* gene as an atherosclerosis modifier gene at the *Ath26* locus (*Han et al., 2019*). However, we observed that even within inbred apoE knockout mice, there was a large coefficient of variation (~50%) in early aortic root lesion area. Since early mouse atherosclerotic lesions are dominated by the accumulation of intimal macrophages (*Nakashima et al., 1994*), we performed an additional AKRxDBA/2 strain intercross to specifically identify candidate genes for bone-marrow-derived macrophage (BMDM) phenotypes, which might also effect atherosclerosis severity. We discovered that ex vivo macrophage phenotypes are generally less variable than atherosclerotic lesion size, which facilitates more robust and precise genetic mapping and therefore faster and more reliable candidate gene

identification. We identified the *Soat1* and *Gpnmb* genes as strong modifier gene candidates for macrophage cholesterol metabolism and lysosome function, respectively; and, we used CRISPR/ Cas9 gene editing to validate them as causal modifier genes (*Hai et al., 2018*; *Robinet et al., 2021*).

In the current study, we assessed IL-1β release from macrophages after inflammasome priming with LPS and activation with ATP; and, we discovered ~two fold more IL-1β was released from DBA/ 2 vs. AKR BMDM. IL-1β is an inflammatory cytokine, and its role in human atherosclerotic disease was proven in the CANTOS trial, where anti-IL-1β monoclonal antibody infusions led to a significant reduction in nonfatal myocardial infarction, stroke, or cardiovascular death (*Ridker et al., 2017*). The goal of this study was to identify and validate candidate genes that contribute to the divergent macrophage inflammasome/IL-1β phenotype between AKR and DBA/2 mice. Additionally, we sought to gain mechanistic insight into how these genes influence this phenotype at a molecular level. Using cryopreserved bone marrow samples from the same $F_4$ strain intercross we utilized in a previous study (*Hai et al., 2018*), we prepared BMDM, primed and activated inflammasomes, and measured IL-1β released into the media. QTL mapping identified a strong locus on the distal end of chromosome 7, which we named inflammatory response modulator 3 (*Irm3*). The *Irm3* QTL interval is located within a larger QTL on chromosome seven that was previously identified in genetic studies from high and low inflammatory responder recombinant partially inbred strains derived from an intercross of eight parental inbred mouse strains, including the DBA/2 strain used in the current study (*Vorraro et al., 2010*). *Irm3* harbors the *Pycard* gene, which codes for the ASC adaptor protein required for assembly of the majority of canonical inflammasomes. Additionally, QTLs on chromosomes 2, 11, and 16 reached statistical significance after correcting for *Irm3*. Here, we describe the identification and validation of *Pycard* as the *Irm3* causal modifier gene, and rs33183533 in the 3' untranslated region (UTR) as the causal genetic variant, which alters *Pycard* mRNA turnover.

## Results

### *Pycard* was identified as a strong candidate gene for the modulation of macrophage IL-1β secretion

DBA/2 vs. AKR BMDM release ~two fold more IL-1β after priming with LPS and subsequent inflammasome activation via ATP treatment (*Figure 1A*). To identify genetic loci responsible for the difference in BMDM IL-1β release (a measure of inflammasome activity) between these strains, QTL mapping was performed. Parental AKR and DBA/2 mice were crossed to generate an $F_1$ population, and their progeny, as well as the progeny of subsequent generations, were brother-sister mated to produce a population of 122 genetically diverse AKRxDBA/2 $F_4$ mice. These $F_4$ mice were genotyped via a dense mouse SNP array, which revealed the desired tapestry of genetic recombination among the cohort (*Hai et al., 2018*), with an average of >2 recombination events per chromosome. BMDM were cultured from these mice and subjected to LPS priming and subsequent ATP treatment. The levels of secreted IL-1β normalized to cellular protein from the $F_4$ BMDM were positively skewed, so a $log_{10}$ transformation was performed to achieve a normal distribution, which is required for subsequent analyses that utilize linear regression models. There was no sex effect on IL-1β levels, thus both sexes were used. QTL mapping was performed to identify regions in the genome where genetic variation was significantly associated with phenotypic variation. A highly significant QTL mapped to distal chromosome 7 (log10 of the odds score (LOD) = 8.60, peak position = 134.80 Mb), which we named inflammatory response modulator 3 (*Irm3*) in accordance with Mouse Genome Informatics nomenclature conventions (*Figure 1B*). Loci on chromosome 2 (147Mb, LOD = 3.79) and chromosome 11 (73Mb, LOD = 3.85) were highly suggestive, falling just short of the genome-wide significance threshold of LOD = 4.02 ($\alpha$ = 0.05) determined by permutation analysis. After correcting for *Irm3* by using its peak marker genotypes as an additive covariate, loci on chromosomes 2, 11, and 16 reached genome-wide significance, and were named *Irm4*, *Irm5*, and *Irm6*, respectively (*Figure 1C*).

*Irm3* has a Bayesian credible interval (probability >0.95 for the causal gene(s) to reside in the interval) of 134.80–138.45 Mb, which contains 66 genes (*Supplementary file 1a*). Separating the $F_4$ BMDM by their genotype at the *Irm3* locus revealed an additive gene dose response for log IL-1β release, with an $R^2$ value of 0.28 (p<0.0001 by ANOVA linear trend test) indicating that this locus is associated

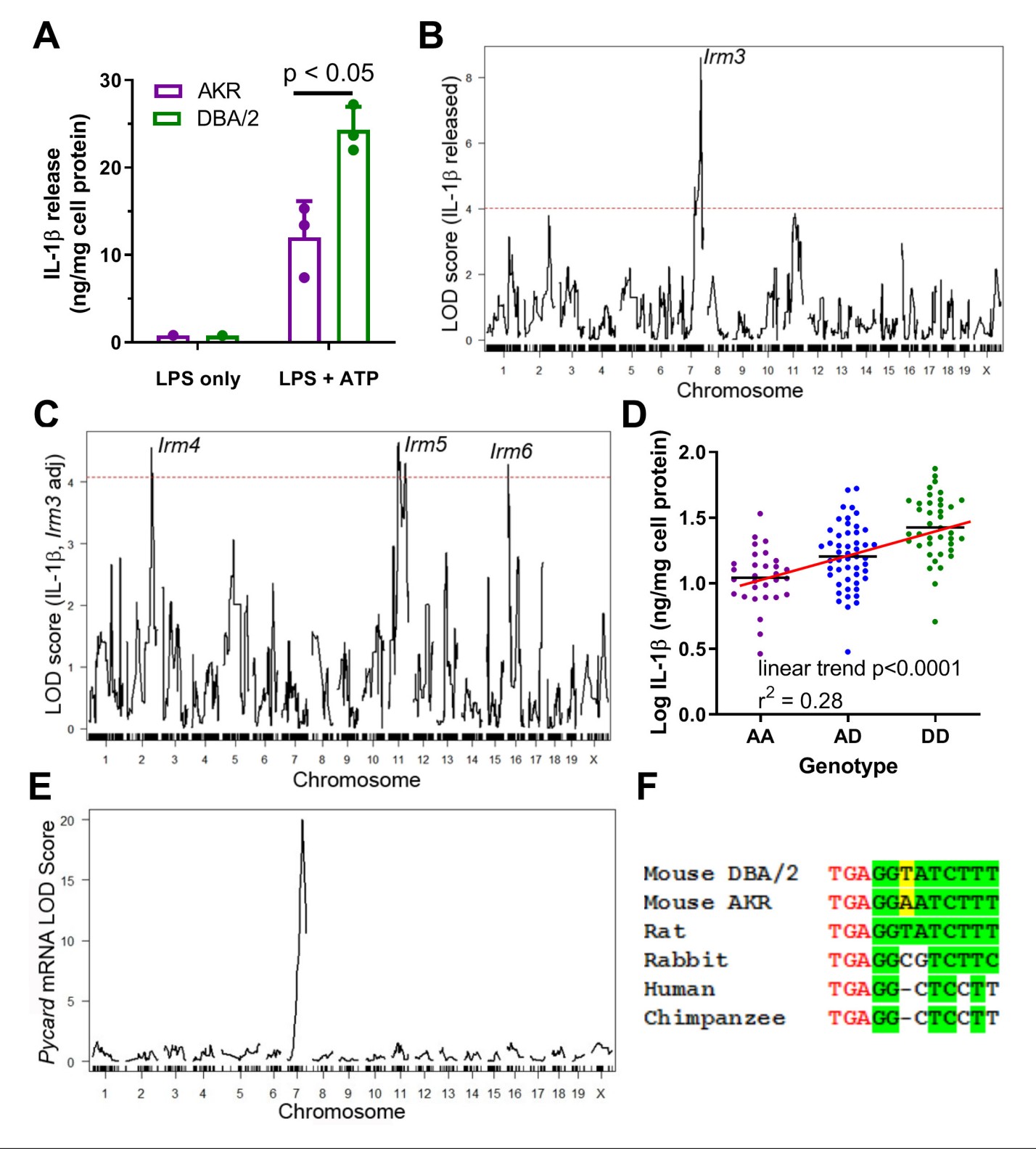

**Figure 1.** Strain effect on IL-1β release associated with Pycard gene. (**A**) IL-1β released into conditioned media (normalized to cell protein) from AKR (magenta) or DBA/2 (green) BMDM after treatment with LPS (4 hr, single well) only or LPS (4 hr)+ATP (30 min, biological triplicates), p<0.05 where indicated by two-tailed t-test, mean and SD shown. Representative of several independent experiments. (**B**) QTL mapping for IL-1β release from $F_4$ BMDMs, showing the prominent *Irm3* QTL peak on chromosome 7. (**C**) QTL mapping for IL-1β release from $F_4$ BMDMs after adjusting for the effect of

*Figure 1 continued on next page*

*Figure 1 continued*

the *Irm3* QTL, showing *Irm4-6* QTLs on chromosomes 2, 11, and 16, respectively (dashed red line in (**B** and **C**) represents the genome-wide significance threshold). (**D**) $Log_{10}$ IL-1β release from $F_4$ BMDMs by genotype at the strongest associated SNP at the *Irm3* peak (AA, AKR homozygotes; AD heterozygotes; DD, DBA/2 homozygotes; $p < 0.0001$ by ANOVA linear trend posttest). (**E**) eQTL mapping for *Pycard* mRNA levels, showing the cis-eQTL peak mapping to chromosome seven where the *Pycard* gene resides. (**F**) *Pycard* gene sequence conservation at the stop codon (red text) and the immediate 3'UTR (conserved residues highlighted in green and the AKR-DBA/2 SNP in yellow). Source data in file ***Source data 1***.

with 28% of the variance in IL-1β release among the $F_4$ BMDM (***Figure 1D***). *Pycard* was selected as the top candidate gene based on its established role in inflammasome assembly, its proximity to the QTL peak (0.33 Mb), and the presence of a strong cis-expression QTL (eQTL, showing genetic variation near that gene is associated with its expression) with a LOD score of 20.0 determined in our prior BMDM transcriptomic study based on an independent intercross of the same two parental strains (***Hsu and Smith, 2013***). Only 3 of the 66 genes in this interval had nonsynonymous SNPs (*Zfp646*, *Bag3*, *Dmbt1*), but none of these SNPs were predicted to alter protein function based on in silico PRO-VEAN analysis (***Choi and Chan, 2015***; ***Supplementary file 1a***). There was one other gene in this interval that we previously found to have a cis-eQTL (*Rgs10*), with a marginal LOD score of 2.6 (2). Taken together, these data suggest that genetic variability within or flanking the *Pycard* gene plays a causal role in manifesting the divergent levels of secreted IL-1β in AKR vs. DBA/2 BMDM, potentially due to differences in *Pycard* gene expression. There is only one SNP (rs33183533) within the *Pycard* gene between the AKR and DBA/2 parental strains, which resides in the 3' UTR. There are also two known upstream SNPs (rs31253258, rs33187231; 5602 and 7736 bp upstream, respectively) and one downstream SNP (rs33182327; 913 bp distal to gene) within 10 kb of the gene. The Sanger Mouse Genomes Project (REL-1505) shows that the 3' UTR SNP rs33183533 DBA/2 allele (T on the coding strand) is the same as the C57BL/6J reference genome allele. Additionally, 24 of the 36 other mouse strains sequenced had a T allele at the 3'UTR SNP, while the AKR allele (A on the coding strand) is shared by 12 of the 36 other mouse strains (***Yalcin et al., 2012***). We performed overlapping PCR (***Supplementary file 1b***) of AKR and DBA/2 genomic DNA covering the entire Pycard gene plus 456 bp and 2361 bp of upstream and downstream flanking region (5310 bp sequenced), which confirmed the presence of the 3' UTR SNP rs33183533 and the downstream SNP rs33182327, with no other sequence variants in *Pycard* exons, introns, or flanking regions. The 3'UTR SNP is 3 bp downstream from the stop codon, and this region is perfectly conserved in rats, which have the DBA/2 allele. Thus, the DBA/2 allele is likely the ancestral mouse allele, but this region is not perfectly conserved in rabbits or primates (***Figure 1E***).

## DBA/2 vs. AKR BMDM have higher *Pycard* expression and form more ASC specks

Real-time quantitative PCR analysis showed that *Pycard* mRNA is expressed at ~2–3 fold higher levels in DBA/2 BMDM relative to AKR BMDM in several independent experiments (***Figure 2A***), in agreement with our prior cis-eQTL data. LPS priming had no effect on *Pycard* mRNA levels or the observed strain difference (***Figure 2A***). Western blotting showed 1.5-fold higher levels of the protein product of *Pycard*, ASC, in DBA/2 vs. AKR BMDM (***Figure 2B***, $p < 0.01$). When inflammasome activation is triggered, ASC assembles into higher-order protein complexes termed 'ASC specks'. Immunostaining for ASC in untreated BMDM showed an expected diffuse cytosolic distribution in both strains (***Figure 2C***). When inflammasomes were primed and activated by treatment with LPS (4 hr) and ATP (30 min), more ASC speck puncta were observed in DBA/2 vs. AKR BMDM (***Figure 2C***). The ASC specks in both strains appeared similar in size and shape, and displayed characteristic perinuclear localization. In a separate experiment, automated image analysis revealed that, after LPS and ATP treatment, 54 AKR BMDM had specks while 8664 did not (0.6% speck positive), and 850 DBA/2 BMDM had specks while 2357 did not (26.5% speck positive, $p < 0.0001$ by Fisher's exact test).

## *Pycard* mRNA half-life is shorter in AKR vs. DBA/2 BMDM

We hypothesized that the *Pycard* 3'UTR SNP could influence *Pycard* mRNA turnover. To determine if *Pycard* mRNA turnover was different between DBA/2 and AKR BMDM, an actinomycin D time course study was performed. The study revealed that *Pycard* mRNA had a longer half-life ($t_{1/2}$) in

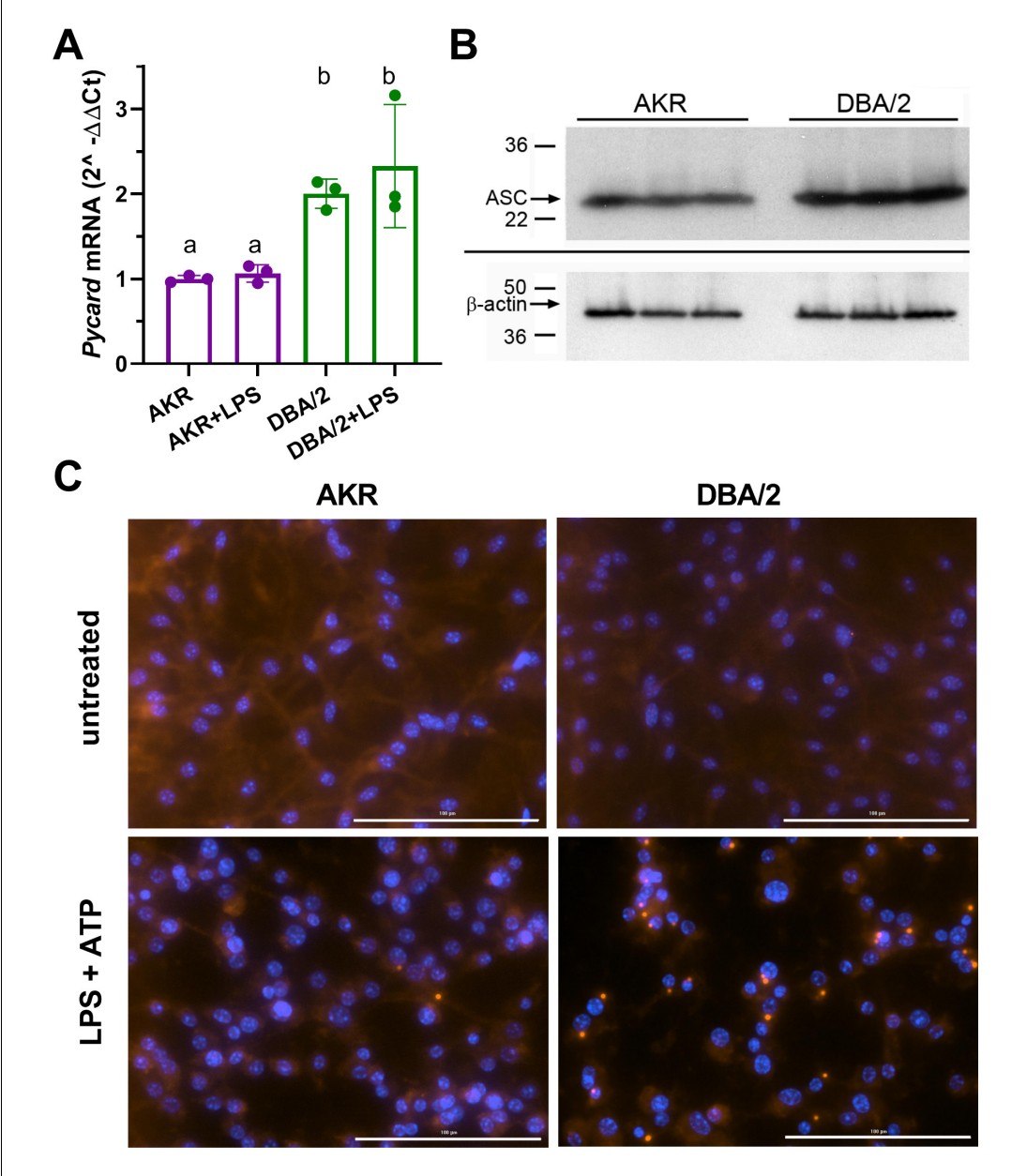

**Figure 2.** Strain effects on *Pycard*/ACS expression and inflammasome speck formation. (**A**) Relative *Pycard* mRNA levels in AKR (magenta) and DBA/2 (green) BMDM, showing no induction by LPS (different letters above columns show p<0.05 by ANOVA Tukey posttest, mean and SD shown). Median of technical triplicates of biological triplicates plotted, representative of two independent experiments. (**B**) Western blot for ASC (top) and β-actin (bottom) in biological triplicate lysates from AKR and DBA/2 BMDM. Densitometric analysis revealed a 50% increase in the ASC/β-actin ratio (p<0.01 by two-tailed t-test). (**C**) Immunofluorescent staining for ACS specks (red) showing assembled inflammasomes and nuclei (blue) in AKR and DBA/2 BMDM with or without inflammasome priming and activation by LPS (4 hr)+ATP (30 min) treatment. For the fields shown in the lower panels, primed and activated AKR BMDM yielded 2 specks among 98 nuclei, while DBA/2 BMDM yielded 23 specks among 58 nuclei. Source data in file *Source data 1*. Unedited western blots in (**B**) unedited western blot source data.docx.

DBA/2 BMDM than in AKR BMDM (1.64 vs. 1.16 hr, respectively, *Figure 3A*), with two-way ANOVA showing significant time (p<0.0001), strain (p=0.031), and interaction effects (p=0.002). Similar results were obtained in an independent experiment. To test for differences in *Pycard* transcription rate, a nuclear run-on experiment was performed, which revealed no significant difference between AKR and DBA/2 BMDM (*Figure 3B*). The secondary structure of a 30 nucleotide sequence of *Pycard* mRNA encompassing the 3'UTR SNP was predicted using the RNAfold WebServer (http://rna.tbi.

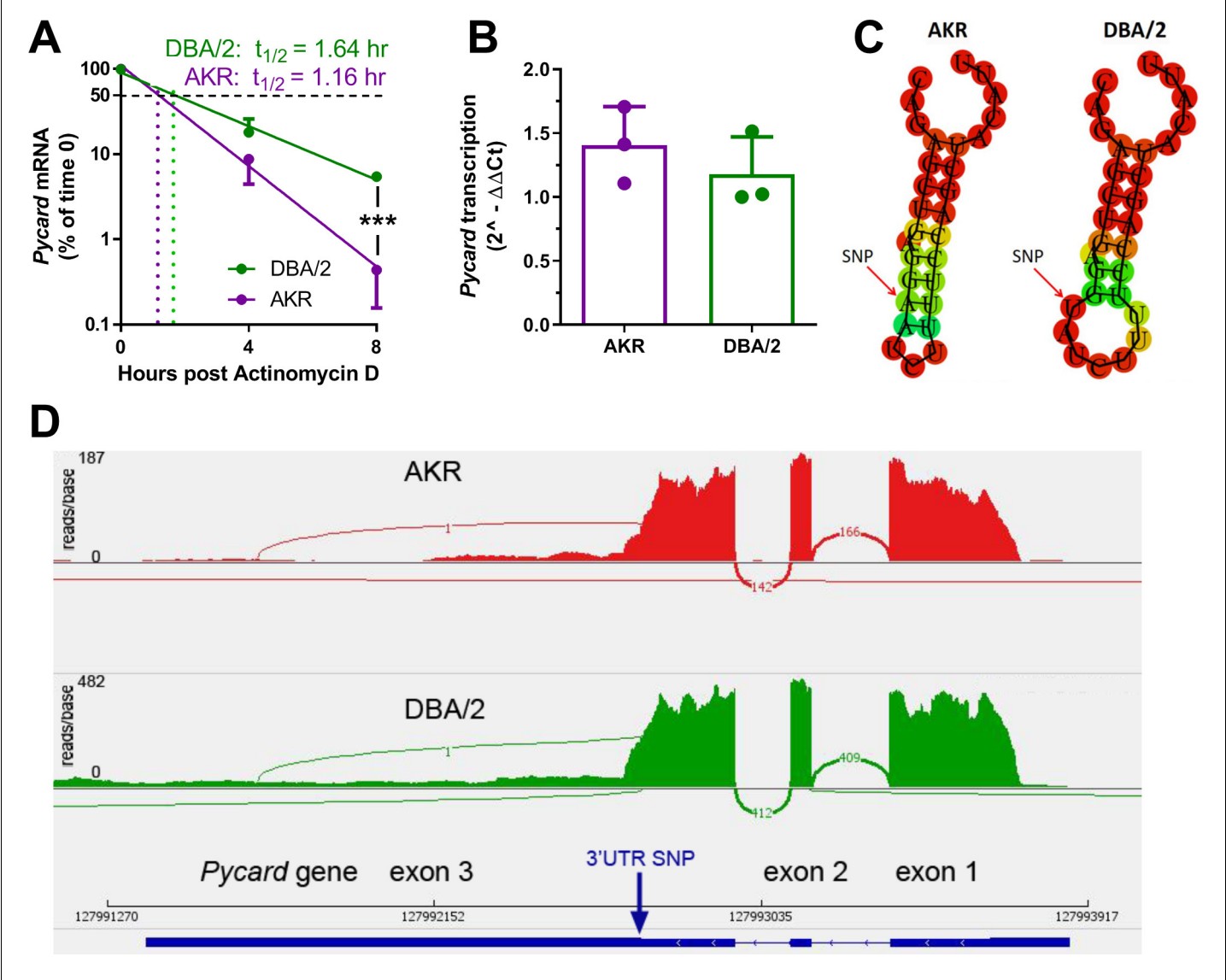

**Figure 3.** Strain effect on *Pycard* mRNA turnover and structure. (A) Semi-log plot of *Pycard* mRNA turnover after Actinomycin D treatment of AKR (magenta) and DBA/2 (green) BMDM (***, p<0.001 by two-tailed t-test). Each point is the mean ± SD of biological triplicates using the mean of technical triplicates. (B) Relative level of *Pycard* mRNA run-on transcription in AKR (magenta) and DBA/2 (green) BMDM (not significant by two-tailed t-test). Biological triplicates, mean and SD shown. (C) Predicted structure of AKR and DBA/2 *Pycard* mRNA segments near the 3'UTR SNP. (D) Sashimi plot of exon junctional reads and read depth histogram (IGV browser view) of RNAseq from AKR and DBA/2 BMDM, with the *Pycard* gene exon-intron structure below (gene on lower strand, 5' to 3' from right to left). Source data in file ***Source data 1***.

univie.ac.at/cgi-bin/RNAfold.cgi). There is a marked difference in predicted mRNA secondary struc-
ture near the SNP, where the AKR allele has a longer stem without a loop after the SNP, and the
DBA/2 allele forms a shorter stem followed by a loop (***Figure 3C***). To determine potential allele-spe-
cific miRNA target sites, we searched the miRDB (***Wong and Wang, 2015***) for miRNAs predicted to
bind to 150 nucleotide segments of *Pycard* mRNA centered on the 3'UTR SNP. Although we identi-
fied the target sequence CCAGCUA, eight nt after the SNP, which is predicted to bind to murine
miR-7688–5 p, miR-7085–3 p, and miR7669-3p, no allele-specific miRNA targets were identified. To
determine if the 3'UTR SNP altered mRNA splicing or isoform expression, we performed RNAseq on
RNA isolated from AKR and DBA/2 BMDM. Assessment of the mapped reads using the IGV browser
(***Robinson et al., 2011***) revealed that both strains express the identical 3-exon isoform (***Figure 3D***),

although both the 5' and the 3'UTRs are shorter than the canonical major transcript isoform (Pycard-201, ENSMUST00000033056.4).

## CRISPR/Cas9 editing of *Pycard* in embryonic stem cells

To confirm the role of the *Pycard* 3'UTR SNP on *Pycard* expression and IL-1β release after inflammasome activation, CRISPR/Cas9 homology directed repair (HDR) gene editing was employed to change the DBA/2 allele (T on coding strand) to the AKR allele (A on coding strand) in the DBA/2J mouse embryonic stem (ES) cell line AC173/GrsrJ. We were able to enrich for HDR editing vs. non-homologous end joining (NHEJ) by using: (1) selection via an HDR-dependent GFP-stop codon reporter; (2) a NHEJ inhibitor; and, (3) cell cycle synchronization (*Figure 4A*). The *Pycard* guide RNA target sequence (on the antisense strand) contained the SNP (*Figure 4B*), so that successful HDR would eliminate the perfect match with the single guide RNA (sgRNA) and limit re-cutting of the edited allele. After co-transfection with sgRNAs and single-strand donor DNAs to correct the GFP stop codon and edit the *Pycard* SNP, we sorted GFP+ cells for clonal growth. We screened 49

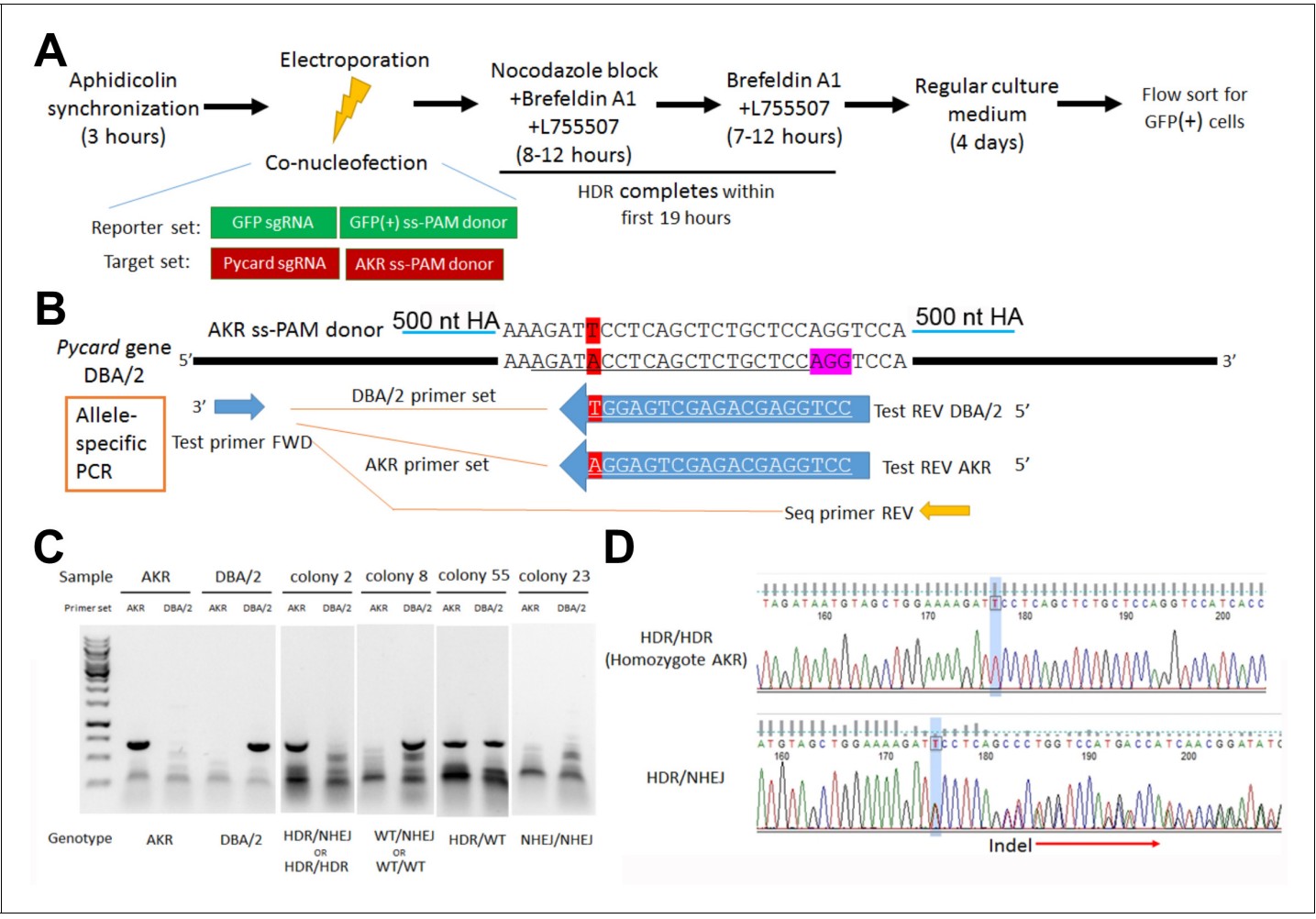

**Figure 4.** *Pycard* gene editing of ES cells to convert the DBA/2 allele to the AKR allele. (**A**) Strategy used to decrease NHEJ by use of HDR reporter and small molecules to modulate cell cycle and inhibit NHEJ. (**B**) Sequence of the AKR allele ss donor (3'UTR SNP highlighted in red) with 500 nt homology arm (HA), which will change the SNP from DBA/2 to AKR and eliminate Cas9 re-cutting, since the SNP is within the sgRNA sequence (underlined in the DBA/2 gene sequence). The sequence of the AKR and DBA/2 allele-specific PCR reverse primers are also shown with the SNP at the 3' end, along with the positions of the common forward primer and the reverse PCR primer used for sequencing the edited clonally derived genomic DNA. (**C**) Example of allele-specific PCR using AKR and DBA/2 genomic controls, and DNA from expanded colonies after gene editing. Genotypes cannot all be distinguished, as one of both alleles may be edited by NHEJ precluding DNA amplification. (**D**) Sanger sequencing of DNA after gene editing showing a homozygous HDR conversion to the AKR allele (top) and a compound heterozygous with editing to one AKR allele and one indel allele due to NHEJ (bottom).

colonies by allele-specific PCR (*Figure 4B,C*) and Sanger sequencing (*Figure 4D*). We obtained five colonies (10.2%) homozygous for the AKR allele, and three of these colonies (H2, H5, and H35) were expanded for functional testing.

## *Pycard* 3'UTR SNP confirmed as a causal modifier via ES cell derived macrophage (ESDM) functional tests

Macrophage-directed differentiation was performed on the three homozygous *Pycard* edited cell lines and their parental clonally derived DBA/2 ES cell line as previously described, and confirmed by acetylated LDL uptake (*Hai et al., 2018*). After LPS and ATP treatment the DBA/2 ESDM released about twofold more IL-1β vs. the three independent *Pycard* edited ESDM lines (p<0.05 or<0.01, *Figure 5A*). qPCR demonstrated that *Pycard* mRNA was ~3 fold higher in DBA/2 ESDM relative to *Pycard* edited ESDM lines (p=0.014, *Figure 5B*). This difference in *Pycard* mRNA was associated with mRNA $t_{1/2}$ of 6.37 vs. 4.56 hr in DBA/2 ESDM vs. *Pycard* edited ESDM, respectively (*Figure 5C*). Western blot analysis showed greater heterogeneity in the levels of ASC protein among the three *Pycard* edited ESDM lines; however, all three edited ESDM lines had reduced ASC protein vs. DBA/2 ESDM (p<0.05 or p<0.001, *Figure 5D*). These results confirm that the *Pycard* 3'UTR SNP is a causal variant that alters *Pycard* mRNA turnover, Pycard mRNA levels, ASC protein levels, and IL-1β release. Of interest, compared to the BMDM, the absolute values of IL-1β secretion and the

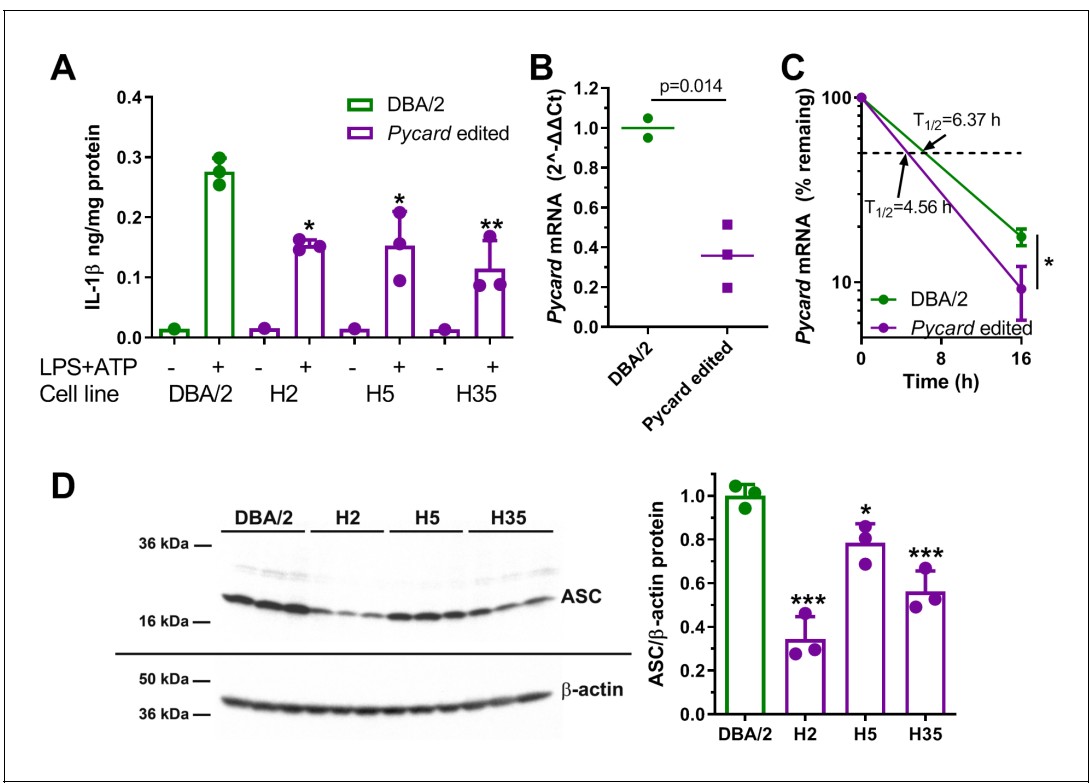

**Figure 5.** Gene editing alters IL-1β release and *Pycard* expression in ESDM. (**A**) IL-1β release from ESDM derived from DBA/2 ES (green) and three independent homozygous *Pycard* edited ES lines (magenta), in the absence or presence of inflammasome priming and activation with LPS +ATP (each point is a biological replicate; *, p<0.05; **, p<0.01 vs DBA/2 derived ESDM in the presence of LPS +ATP by ANOVA with Dunnett's multiple comparisons test, mean and SD shown). (**B**) *Pycard* mRNA levels in two DBA/2 ESDM differentiations and three independent homozygous *Pycard* edited ESDM (p=0.014 by two-tailed t-test). Each point is a biological replicate of qPCR technical triplicates, mean shown. (**C**) Semi-log plot of *Pycard* mRNA turnover after Actinomycin D treatment of ESDM derived from two differentiations of DBA/2 ES and three independent homozygous *Pycard* edited ES lines (magenta), (*, p<0.05 by two-tailed t-test). Each point is the average of biological triplicates of qPCR technical triplicates, mean and SD shown. (**D**) Left side, western blot for ASC (top) and β-actin (bottom) from ESDM lysates derived from DBA/2 ES and three independent homozygous *Pycard* edited ES lines. Right side, densitometry of western blot showing ASC levels are lower in all three *Pycard* edited cell lines (*, p<0.05; ***, p<0.001 vs. DBA/2 derived ESDM by ANOVA with Dunnett's multiple comparisons test, mean and SD shown).

*Pycard* mRNA $t_{1/2}$ were different in the ESDM, suggesting these macrophages may be less mature than BMDM.

## Discussion

QTL mapping of macrophage IL-1β secretion performed on an AKRxDBA/2 $F_4$ population identified an approximate 3.5 Mb locus on distal chromosome 7, which we named *Irm3*. *Pycard* resides in *Irm3* and was recognized as a very strong candidate gene because its protein product, ASC, is an inflammasome adaptor protein, and *Pycard* had a strong cis-eQTL in a prior study (*Hsu and Smith, 2013*). DBA/2 vs. AKR BMDM were shown to express *Pycard* and ASC at higher levels and assemble more ASC specks upon LPS and ATP stimulation. The lone SNP in the *Pycard* gene sequence (rs33183533) between AKR and DBA/2 mice, located in the 3′UTR, was unequivocally verified as a significant modulator of macrophage IL-1β secretion and *Pycard* expression using CRISPR/Cas9 gene editing.

Another group previously identified the inflammatory response modulator 1 (*Irm1*) QTL on the distal end of mouse chromosome 7, which was associated with both IL-1β release from ex vivo stimulated blood leukocytes and in vivo leukocyte Biogel infiltration. This QTL was mapped by intercrossing incompletely inbred high and low responder strains derived from an eight-strain intercross followed by selective breeding (*Vorraro et al., 2010*). The *Irm1* 1-LOD confidence interval for leukocyte IL-1β release extends from 130.75 to 144.75 Mb, which encompasses the *Irm3* QTL (134.80–138.45 Mb) discovered in the current study. Therefore, it is possible that *Irm3* represents a fine-mapping of a candidate gene in the *Irm1* locus, which is plausible considering we started with inbred strains, created an $F_4$ versus an $F_2$ cross, and used a denser SNP panel. Although these investigators never identified the gene responsible for *Irm1*, they ruled out *Pycard* as the causal gene for *Irm1* due to equal distribution of the 3′UTR SNP we evaluated (rs33183533) and another SNP in the first intron of the *Pycard* gene (rs51540238) among the high and low responder parental strains used in their study (*Vorraro et al., 2010*). However, their study did not examine the *Pycard* 3′UTR SNP genotype effect on IL-1β produced in the $F_2$ mice, and thus it is possible that the *Irm1* QTL is indeed due to the *Pycard* 3′UTR SNP due to the selection and success of specific breeders within the incompletely inbred parental high and low responder strains. In addition, *Pycard* mRNA and ASC expression levels in the $F_2$ mice were not reported in their study. As with any QTL study, it is also possible that there are additional genes that modulate IL-1β secretion at the distal end of chromosome 7, and another gene is the primary modulator for the strains used in their study. Additional mouse QTLs that overlap *Irm3* include *Lsq-1*, a QTL for hindlimb ischemia (*Dokun et al., 2008*), and *Civq1*, a QTL for infarct volume following ischemic stroke (*Keum and Marchuk, 2009*). Neither study postulates a role for *Pycard* in these phenotypes.

Human GWAS studies revealed an SNP near *Pycard* that is associated with bronchodilator response in asthma, although this association does not meet the commonly used threshold ($<5\times10^{-8}$) for genome-wide significance (*Lutz et al., 2015*). The dbSNP database lists only two common human SNPs within the *PYCARD* gene, rs115908198 located in the 3′UTR and rs73532217 located in the first intron. These two SNPs are only common (>1% minor allele frequency) in African populations with minor allele frequencies of ~7% and ~2%, respectively. However, neither of these SNPs have been studied mechanistically to determine effects on PYCARD expression. The human GTEx portal (Release V8) study of gene expression in many postmortem human tissues identifies 80 distinct cis-eQTL SNPs near the *PYCARD* gene associated with *PYCARD* mRNA expression at $p<5\times10^{-8}$, including the intronic SNP rs73532217. Due to linkage disequilibrium, the co-inheritance of nearby variants from ancestral chromosomes, it is likely that only one or a few of these cis-eQTL SNPs are true regulatory SNPs that alter mRNA production or turnover, which must be confirmed by mechanistic investigations such as those performed in the current study.

Searching the human GWAS catalog (https://www.ebi.ac.uk/gwas/home) for 'interleukin-1 beta measurement' in January, 2021 yielded four studies; however, three of these found no common SNPs associated with plasma/serum IL-1β at genome-wide significance, albeit these studies did identify SNPs associated with other cytokines (*Matteini et al., 2014*; *Ahola-Olli et al., 2017*; *Offenbacher et al., 2018*). The single study that identified common SNPs associated with plasma IL-1β found two independent SNPs on chromosome 6, near the HLA locus, associated with IL-1β levels (*Sliz et al., 2019*). We suspect that human GWAS studies for IL-1β may be susceptible to false negative findings, as IL-1β levels have a large environmental component due to its response to infection

or inflammation. Therefore, mouse studies may be useful to identify genes, pathways, and mechanisms that regulate IL-1β release after inflammasome activation, which would be difficult to perform in humans. Performing in vitro human genetic studies with a large panel of human-induced pluripotent stem cells differentiated into macrophages or dendritic cells (*Warren and Cowan, 2018*) might be an excellent alternative to identify common human genetic variants associated with inflammasome priming/activation and IL-1β release.

ASC protein/*PYCARD* gene has been previously shown to be subject to additional regulatory control by post-translational modifications such as ubiquitination and phosphorylation and by mRNA splicing (*Hoss et al., 2017*). The tyrosine kinase Syk is required for Nlrp3 inflammasome activation in mouse BMDM (*Gross et al., 2009*). Both Syk and Jnk kinases are required for Nlrp3 and AIM2 inflammasome activation in mouse and transfected human cells, which are associated with ASC phosphorylation and oligomerization (*Hara et al., 2013*). Human ASC tyrosines 146 and 187 are critical for Syk-mediated ASC oligomerization and inflammasome activity (*Lin et al., 2015*). Syk phosphorylates Pyk2 kinase, which then directly phosphorylates ASC on tyrosine 146 (*Chung et al., 2016*). In contrast IκB kinase α (IKKα) binds directly to ASC, requiring ASC serines 16 and 193, and negatively regulates Nlrp3 inflammasome activity, so that upon inflammasome activation by ATP or other agonists IKKα is released from ASC (*Martin et al., 2014*). Protein phosphatases also regulate ASC oligomerization and activity, with the broad-spectrum tyrosine phosphatase inhibitor phenylarsine oxide reducing ASC assembly and speck formation (*Mambwe et al., 2019*), although this appears to be in the opposite direction of the Syk effects on ASC phosphorylation. The protein tyrosine phosphatase PTPN2 is a negative regulator of inflammasome activity, as demonstrated in myeloid-specific Ptpn2 knockout mice that have increased inflammasome activity and ASC phosphorylation via increased Jnk activity (*Spalinger et al., 2018*). Based on cDNA sequences, RT-PCR, and ASC domain-specific antibodies, alternative splicing of PYCARD mRNA has been proposed leading to three alternative human ASC isoforms; and, transfection overexpression studies found some inflammasome activity with the alternative exon two skipped isoform (*Bryan et al., 2010*).

Our study identified an additional ASC regulatory mechanism in mice due to a *Pycard* 3′ UTR SNP, where the allele that decreases *Pycard* mRNA stability was found in ~1/3 of sequenced mouse strains. We further showed that the *Pycard* 3′UTR SNP led to different predicted mRNA secondary structure without altering the transcript isoform expressed; and, we identified three mouse miRNAs with target sequences eight nt downstream of the 3′UTR SNP. Additional studies would be required to determine if the different structures of AKR and DBA/2 *Pycard* mRNAs influence the binding of miRNAs or putative RNA-binding proteins that might alter transcript turnover.

## Materials and methods

### Key resources table

| Reagent type (species) or resource | Designation | Source or reference | Identifiers | Additional information |
|---|---|---|---|---|
| Gene (*Mus musculus*) | *Pycard* | Ensemble | ENSG00000103490 | |
| Strain, strain background (*Mus musculus*) | AKR/J | JAX | 648 | |
| Strain, strain background (*Mus musculus*) | DBA/2J | JAX | 671 | |
| Cell line (*Mus musculus*) | DBA/2J mouse ES cell line AC173/GrsrJ | JAX | 000671C02 | |
| Cell line (*Mus musculus*) | (Puromycin-resistant MEF feeder cells) | Cell Biolabs | CBA-312 | |

*Continued on next page*

*Continued*

| Reagent type (species) or resource | Designation | Source or reference | Identifiers | Additional information |
|---|---|---|---|---|
| Cell line (*Mus musculus*) | Neomycin-resistant MEF feeder cells (Cell Biolabs, CBA-311) | Cell Biolabs | CBA-311 | |
| Transfected construct (*S. pyogenes*) | Cas9 expression plasmid pSpCas9(BB)−2A-Puro | Addgene | PX459 | |
| Transfected construct (*Aequorea Victoria*) | MSCV-miRE-shRNA IFT88-PGK-neo-IRES-GFP plasmid, | Addgene | 73576 | |
| Antibody | Mouse monoclonal anti Cas9 (Diagenode, C15200203) | Diagenode | C15200203 | IHC (1:1000) |
| Antibody | Rabbit monoclonal anti-mouse AS | Cell Signalling | 67824 | WB(1:500) |
| Antibody | Rabbit polyclonal anti-ASC N-terminus | Santa Cruz Biotech | Sc-22514-R | IF (10 µg/ml) |
| Antibody | Goat polyclonal alexa Flour 568 anti-rabbit IgG | ThermoFisher | A-11011 | IF (2 µg/ml) |
| Antibody | Mouse monoclonalHRP-conjugated anti β-actin | Santa Cruz Biotech | Sc-47778-HRP | (WB (1:20,000)) |
| Sequence-based reagent | qPCR for mouse Pycard | ThermoFisher | 4331182, Assay ID: Mm00445747_g1 | |
| Sequence-based reagent | qPCR for mouse Actb | ThermoFisher | 4448484, Assay ID: Mm02619580_g1 | |
| Sequence-based reagent | Nuclear run-on qPCR for mouse Pycard | ThermoFisher | 4441114, Assay ID: AJMSHN7 | |
| Sequence-based reagent | sgRNA for GFP stop codon | Synthego | Custom | GGGCGAGGGCGAUGCCACCU |
| Sequence-based reagent | sgRNA for Pycard 3' UTR | Synthego | Custom | AGAUACCUCAGCUCUGCUCC |
| Peptide, recombinant protein | Leukaemia inhibitory factor | Millipore-Sigma | ESG1107 | |
| Peptide, recombinant protein | Mouse IL-3 | R and D Systems | 403 ML | |
| Commercial assay or kit | Mouse Il-1β ELISA | R and D Systems | MLB00C | |
| Commercial assay or kit | SuperScript VILO cDNA synthesis kit | ThermoFisher | 1175505 | |
| Commercial assay or kit | iScript cDNA synthesis kit | BioRad | 1708891 | |
| Chemical compound, drug | LPS from *E. coli* O55:B5 | Sigma | L6529 | |

*Continued on next page*

*Continued*

| Reagent type (species) or resource | Designation | Source or reference | Identifiers | Additional information |
|---|---|---|---|---|
| Chemical compound, drug | ATP | Sigma | A2383 | |
| Software, algorithm | Custom special R functions | This paper | https://github.com/BrianRitchey/qtl | See Methods section |
| Software, algorithm | r/QTL | Reference 30 in this paper | https://rqtl.org/download/ | |
| Other | 30 µm sterile filters | Sysmex | 04-004-2326 | To filter out embryonic bodies during ESDM differentiation |

## Generation and genotyping of AKRxDBA/2 F$_4$ mice

All animal studies were approved by the Cleveland Clinic Institutional Animal Care and Use Committee. Parental wild type male AKR/J and female DBA/2J mice, obtained from JAX (# 648 and 671), were crossed to create the F$_1$ generation, fixing the Y chromosome from the AKR strain. Two breeding pairs of F$_1$ mice were bred to generate F$_2$ mice, and two breeding pairs of F$_2$ mice were used to generate F$_3$ mice. Six breeding pairs of F$_3$ mice were used to generate the 122 F$_4$ mice, which consisted of 70 males and 52 females. Healthy F$_4$ mice were sacrificed at 8–10 weeks of age. Ear tissue was collected from each mouse and digested overnight at 55°C in lysis buffer containing 20 mg/mL proteinase K. DNA was ethanol precipitated and resuspended in 10 mM Tris 1 mM EDTA (pH = 8). Femurs were promptly flushed after sacrifice, and resultant bone marrow cells were washed, aliquoted, and cryopreserved. Cells were thawed and differentiated into macrophages at the time of experimentation, as described below. F$_4$ mice were genotyped as described previously (*Hai et al., 2018*). Briefly, the GeneSeek MegaMUGA SNP array was used, and filtering for call frequency and strain polymorphism using parental and F$_1$ DNA yielded 16,975 informative SNPs that were used for QTL analysis. All marker locations are based on NCBI Mouse Genome Build 37.

## Bone-marrow-derived macrophages

Cryopreserved bone marrow cells were resuspended and plated in macrophage growth medium (DMEM, 10% FBS, 20% L-cell conditioned media as a source of Macrophage Colony Stimulating Factor). Media was renewed twice per week. Cells were used for experiments 11–14 days after plating, when the cells were confluent and fully differentiated into BMDM. Three of the 122 F$_4$ frozen bone marrow cells did not generate macrophages yielding 119 samples assessed below.

## IL-1β release assay

BMDM or ESDM were primed with 1 µg/mL LPS from *Escherichia coli* O55:B5 (Sigma, L6529) for 4 hr at 37°C and subsequently treated with 5 mM adenosine triphosphate (ATP) (Sigma; A2383) for 30 min at 37°C. Media were collected and briefly centrifuged to pellet any cellular debris and the resultant supernatant was collected. IL-1β levels were measured via a mouse IL-1β ELISA assay according to the manufacturer's instructions (R and D Systems, MLB00C). Released IL-1β levels were normalized to cellular protein, as determined by the bicinchoninic acid protein assay (ThermoFisher, 23227) of total cell lysates prepared by incubation at 37°C for $\geq$4 hr in 0.2 N NaOH, 0.2% SDS.

## Quantitative trait locus (QTL) mapping of macrophage IL-1β release

QTL mapping of $\log_{10}$ IL-1β released from 119 AKR x DBA/2 F4 BMDM was performed using R/qtl software (*Broman et al., 2003*). The 'scanone' function was utilized using Haley-Knott regression by specifying the 'method' argument as 'hk'. False discovery rates (FDRs) were estimated via permutation analysis, using 10,000 permutations by specifying the 'n.perm' argument in the 'scanone' function. QTL credible intervals were determined using the Bayesian credible interval ('bayesint')

function in R/qtl, with the 'prob' argument set at 0.95. QTL mapping for *Irm4-6* was performed using the genotypes from the most strongly associated *Irm3* marker as an additive covariate ('addcovar') in the 'scanone' function of R/qtl, and again subjected to 10,000 permutation analyses to determine FDRs. Loci that reached significance were designated *Irm4-6* based on their occurrence scanning left to right across the genome. To aid in prioritizing candidate genes, a custom R function termed 'flank_LOD' was written (https://github.com/BrianRitchey/qtl/blob/master/flank_LOD.R). 'flank_LOD' utilizes the 'find.flanking' function in R/qtl and returns the LOD score of the nearest flanking marker for a given candidate gene position based on 'scanone' output data.

## Pycard mRNA expression assay

RNA was extracted from BMDM or ESDM by scrapping cells in QIAzol reagent and homogenization by 5–10 passages through a 27 gauge syringe with subsequent phenol/chloroform extraction. RNA was purified using the miRNeasy Mini Kit (Qiagen, 217004) with on-column DNA digestion according to manufacturer's instructions. cDNA was generated using SuperScript VILO Master Mix (Thermo-Fisher, 11755050) or IScript cDNA Synthesis Kit (BioRad, 1708891). mRNA levels were determined via TaqMan qPCR assays for mouse *Pycard* (ThermoFisher, 4331182, Assay ID: Mm00445747_g1), with *Actb* (4448484, Assay ID: Mm02619580_g1) serving as an internal control. Samples were run for 40 cycles on an Applied Biosystems StepOnePlus Real-Time PCR System using the comparative Ct method. Resultant data were analyzed using the $2^{-\Delta\Delta Ct}$ method relative to the average AKR ΔCt for BMDM, and the average DBA/2 ESDM ΔCt for ESDM. In some experiments, *Pycard* mRNA turnover was assessed by treating cells with 10 μg/mL Actinomycin D (Sigma, A1410) at various time points before cells were harvested.

## Nuclear run-on

Previously published methods were followed (*Roberts et al., 2015*; *Patrone et al., 2000*). AKR and DBA/2 BMDM were first primed with 1 μg/mL LPS for 4 hr. Cells were collected into strain-specific pools, counted on a hemocytometer, and lysed in NP-40 lysis buffer to obtain nuclei. In vitro RNA synthesis with biotin-16-UTP was performed using 50 million nuclei per reaction, with triplicate concurrent reactions for each strain. Reactions were incubated at 30℃ for 30 min, and RNA was extracted and purified using the miRNeasy Mini Kit. Newly synthesized transcripts were selected for biotin-16-UTP incorporation using streptavidin coated magnetic beads (ThermoFisher, Dynabeads M-280). Beads were extracted using QIAzol reagent, with subsequent phenol/chloroform extraction, and RNA was ultimately isopropanol precipitated with glycogen added as a carrier. cDNA was generated using SuperScript VILO Master Mix, and RNA levels were determined via TaqMan qPCR assays for mouse *Pycard* (ThermoFisher, 4441114, Assay ID: AJMSHN7), with *Actb* (ThermoFisher, 4448484, Assay ID: Mm02619580_g1) serving as an internal control. A custom TaqMan assay was designed for *Pycard*, with primers spanning an intron-exon boundary. The specific *Actb* TaqMan assay was selected because primers were within a single exon. These assays comply with the primer design guidelines for nascent transcript quantification as previously described (*Roberts et al., 2015*).

## ASC western blot assay

Proteins were extracted from BMDM or ESDM in triplicate with RIPA buffer (Pierce, 89900) as previously described (*Hai et al., 2018*). Fifteen to 50 μg of each cell lysate was mixed with SDS sample buffer and incubated for 8 min at 95℃ then immediately cooled on ice for 8 min. Proteins were separated by SDS-PAGE (ThermoFisher, XP04200BOX) for 2 hr at 110V, and then transferred to a PDVF membrane. After incubating with Casein Blocker in TBS (ThermoFisher, 37532) for 1 hr at room temperature, the membrane was incubated overnight at 4℃ with primary rabbit ASC antibody (Cell Signaling, 67824) 1:500 in blocking buffer. After washing with PBS-0.05% Tween20, the membrane was probed with HRP-conjugated secondary antibody (goat anti-rabbit) 1:20,000 in blocking buffer for 1 hr at room temperature. The bands were visualized by HRP chemiluminescence detection Reagent (Millipore, WBKLS0500). The membrane was re-probed with HRP-conjugated anti β-actin (Santa Cruz Biotech, sc-47778-HRP) 1:20,000 in blocking buffer for 1 hr at room temperature and visualized in the same way. Densitometric analysis of bands was performed using ImageJ software.

## ASC speck imaging and quantification

BMDM grown in 24-well plates were fixed in ethanol and blocked in 1% BSA. Immunostaining against ASC was performed using ASC (N-15) antibody (Santa Cruz Biotechnology; sc-22514-R) at 10 µg/mL for 1 hr at room temperature. Alexa Fluor 568 anti-rabbit (ThermoFisher; A-11011) was then incubated at 2 µg/mL for 1 hr at room temperature. DAPI (Simga; D9542) staining (300 nM) was performed for five minutes at room temperature. Images were captured using the Cytation 3 Cell Imaging Multi-Mode Reader (Biotek) using a ×20 objective lens. For one experiment, automated images were captured using the Cytation three instrument from triplicate LPS +ATP treated BMDM wells per strain using a 12 × 10 grid in each well (120 total images per well). Nuclei and ASC specks were counted using the cellular analysis feature in Gen5 software (Biotek). Nuclei and speck counts were compared in AKR vs. DBA/2 BMDM by Fisher's exact test contingency table analysis.

## RNA sequencing

Total RNA was prepared from AKR and DBA/2 BMDM using miRNeasy Mini Kit (Qiagen; # ID 217004) with on-column DNA digestion according to manufacturer's instructions. RNA integrity and RNAseq was performed by the University of Chicago Genomics Core. 30 million paired end reads were obtained using the Illumina NovaSeq 6000 with library preparation using the oligo dT directional method. Fasta files were preprocessed, aligned and quantified using the nf-core/rnaseq pipeline version 1.4.2, which is part of nf-core framework for community-curated bioinformatics pipelines (*Ewels et al., 2020*). Specifically, reads were aligned using the STAR aligner version 2.6.1d with the Gencode M25 transcriptome and GRCm38 primary assembly genome and all default values of the pipeline were used except read trimming was not performed before alignment. The BAM and BAM index files were viewed and Sashimi plots prepared using the Integrative Genomics Viewer (IGV) browser (*Robinson et al., 2011*).

## Cell lines and cell culture

Puromycin-resistant MEF feeder cells (Cell Biolabs, CBA-312) and neomycin-resistant MEF feeder cells (Cell Biolabs, CBA-311) were cultured in DMEM high glucose supplemented with 10% fetal bovine serum and 1% PenStrep at 37°C, then inactivated with 10 µg/ml mitomycin C (Sigma, M4287) for 2 hr for mouse embryonic stem cell culture, as previously described (*Hai et al., 2018*). DBA/2J mouse ES cell line AC173/GrsrJ (JAX, 000671C02), was cultured on 0.1% gelatin-coated plates with mitomycin C inactivated MEFs, in ES culture medium (DMEM high glucose with 15% fetal bovine serum, 1% MEM Non-Essential Amino Acids, 1% PenStrep, 0.1 mM 2-mercaptoethanol, $10^3$ unit/ml leukemia inhibitory factor (Millipore Sigma, ESG1107), 1 µM PD0325901 (Sigma, PZ0162) and 3 µM CHIR99021 (Sigma, 361571)), at 37°C. All cell lines were detached with trypsin and frozen with 80% ES culture medium supplemented with 10% DMSO and an additional 10% FBS. Only mouse cell lines used. No STR profiling methods are available to authenticate mouse cell lines. DNA sequencing was used to confirm editing of the DBA/2 allele to the AKR allele of the mouse ES cell line used. Mycoplasm testing was negative.

## Gene editing by homology directed repair (HDR)

CRISPR/Cas9 HDR was employed in DBA/2 ES to make a single base pair change in the *Pycard* 3'UTR SNP from the DBA/2 allele to the AKR allele. In order to enhance the low frequency of HDR, multiple strategies were employed, including reporter-dependent co-selection, non-homologous end joining (NHEJ) inhibition, and cell cycle control (*Figure 4A,B*). Five µg Cas9 expression plasmid pSpCas9(BB)−2A-Puro (Addgene, PX459) was stably transfected into 8 × $10^5$ DBA/2 ES cells via electroporation, using a Lonza Amaxa nucleofector II with program A-24 and mouse ES cell nucleofector kit (Amaxa, VAPH-1001). Transfected cells were plated in 2 mg/ml puromycin in ES culture medium on the puromycin resistant MEFs in P100 tissue culture dishes. One week later, the medium was replaced with regular ES culture medium. Three to 7 days later individual colonies were picked and expanded and western blot was used to confirm Cas9 protein expression (Diagenode, C15200203). Cas9 expression in a high expressing line was confirmed by immunohistochemistry (*Hai et al., 2018*).

To create a selectable HDR reporter, we used site-directed mutagenesis of a GFP expression plasmid (MSCV-miRE-shRNA IFT88-PGK-neo-IRES-GFP plasmid, Addgene # 73576), and substituted

a single G for a C to introduce an in frame stop codon (TAG) in place of a tyrosine codon (TAC) in the initial region of the GFP coding sequence and simultaneously generate a new PAM sequence (AGG) from the original sequence (ACG). Two oligos (F: GCGATGCCACCTAGGGCAAGCTGACCC TG and R: CAGGGTCAGCTTGCCCTAGGTGGCATCGC) were used with the QuikChange II mutagenesis kit (Agilent, 200523). We confirmed that this mutation (GFPstop) extinguished GFP expression compared to the parent plasmid by transient transfection. Two µg of the GFPstop expression plasmid was stably transfected into the Cas9 stably transfected DBA/2 ES cell line as described above, followed by selection in growth media containing 900 µg/ml G418. Colonies were expanded and stable transfection confirmed by PCR of genomic DNA using GFP specific primers (F: A TAAGGCCGGTGTGCGTTTGTCTA; R: CGCGCTTCTCGTTGGGGTCTTTG).

sgRNAs were designed to target Cas9 nuclease to the GFPstop (GGGCGAGGGCGAUGCCACC U) and the *Pycard* 3' UTR SNP (AGAUACCUCAGCUCUGCUCC) using ZiFit software (*Sander et al., 2010*) and purchased from Synthego with their chemical modification to increase stability. To perform HDR gene editing we prepared two ssDNA donor templates by PCR and single strand degradation as described below. The GFP donor repairs the created stop codon and removes the PAM sequence utilized by the GFP sgRNA, thus HDR would simultaneously generate GFP + cells that cannot be recut by Cas9 nuclease. The GFP donor PCR primers 5' phosphorylated-CGCGCTTCTCG TTGGGGTCTTTG (the non-PAM strand) and ATAAGGCCGGTGTGCGTTTGTCTA (PAM strand) generated an 1188 bp dsDNA product using the non-mutated GFP expression vector as a template. The Guide-it Long ssDNA Production System (Takara Bio, 632644,) was used to degrade the 5' phosphorylated strand to generate the ssDNA PAM-strand GFP donor. The *Pycard* donor creates the AKR allele at the 3' UTR SNP and changes the sgRNA target sequence such that HDR would simultaneously generate the AKR allele that cannot be recut by Cas9 nuclease. The *Pycard* donor PCR primers 5' phosphorylated -TGTGTCCCCTTGTTCGTCTACCC (non-PAM strand) and TTTC TAAGCCCCATTGCCTGTTTT (PAM strand) generated an 1144 bp dsDNA product using AKR mouse genomic DNA as a template. The ssDNA PAM-strand *Pycard* donor was generated as described above.

Electroporation for HDR was performed as described above on $2 \times 10^6$ Cas9+/GFPstop stably transfected ES cells using 100 pmol each of the GFP and *Pycard* template donors and 2 µg of each sgRNA. In order to increase HDR we used cell cycle control, as previously described (*Lin et al., 2014*), by treatment of 70% confluent ES cells for 3 hr with 5 µg/ml aphidicolin prior to the electroporation and 8 hr treatment with 300 ng/ml nocodazole after the electroporation. We also inhibited NHEJ (*Yu et al., 2015*) by adding 0.1 µM brefeldin A1 and 5mM L755507 for the first 16 hr after electroporation and then replacing with ES growth medium. Ninety-six hr after electroporation, the transfected cells were detached for fluorescent activated cell sorting (FACS) to collect the GFP + cells competent for HDR. GFP + cells were plated at low density on inactivated MEFs for 10 days and individual colonies were picked and expanded.

Fifty-five clonally derived GFP⁺ cell lines were subjected to *Pycard* genotyping by allele-specific PCR. Genomic DNA was extracted from each cell line and used as a PCR template in separate reaction with AKR allele-specific primers (F: AACAGCCCCACCCCCAAAATCCAC; R: CCTGGAGCA-GAGCTGAGGA) and DBA/2 allele-specific primer (F: AACAGCCCCACCCCCAAAATCCAC; R: CC TGGAGCAGAGCTGAGGT), which only differed from each other in the 3' terminal nucleotide on the R primer (*Figure 4B*). Genomic DNA from AKR and DBA/2 mice was used as positive and negative controls for the respective allele-specific PCR reactions. The allele specific PCR reactions can also yield a false negative if an indel is introduced by NHEJ such that neither primer pair would work. Thus, samples yielding product only with the DBA/2 primers can be derived from unedited wild-type DBA/2 (WT) WT/WT alleles, or WT/NHEJ alleles (*Figure 4C*). Likewise, for samples yielding product only with the AKR primer set can be derived from homozygous AKR alleles derived by HDR, HDR/ HDR alleles, or HDR/NHEJ alleles. To distinguish HDR/HDR from HDR/NHEJ genotypes we performed a non-allele-specific PCR reaction (F: AACAGCCCCACCCCCAAAATCCAC; R: GTGGC TTTCCTTGATTCT) for sequencing (*Figure 4B*). The PCR product was purified with ExoSAP-IT PCR Product Cleanup Reagent (ThermoFisher), and Sanger sequenced using the primer CATAACTTGGG TCTGTGG. If only one sequence was obtained corresponding to the HDR allele the genotype is homozygous HDR/HDR, that is mutated to the AKR allele at the *Pycard* 3' UTR SNP (*Figure 4D*). In subsequent functional studies, three independent homozygous (H) HDR/HDR *Pycard* edited cell lines were used, named H2, H5, and H35.

## ESDM differentiation

The macrophage differentiation protocol was adapted from previous publications (*Zhuang et al., 2012*; *Yeung et al., 2015*). Three homozygous *Pycard* edited cell lines and their parental Cas9+/ GFPstop stably transfected DBA/2 ES cell line (DBA/2) were cultured on inactivated MEFs as described above. These cell lines were passaged without inactivated MEFs for two generations from low density to 80% confluence to decrease MEF contamination. To eliminate any possible residual MEFs, detached cells were bound to gelatin coated tissue culture plates in ES culture medium at 37° C for 30 min, such that MEFs stuck to the plate and ES cells remained in the supernatant. $6 \times 10^5$ ES cells from each cell line were resuspended in macrophage differentiation medium (MDM), which consists of DMEM high glucose, 15% FBS, 1% PenStrep, 1% MEM non-essential amino acids, 0.1 mM 2-mercaptoethanol, 3 ng/ml mouse IL-3 (R and D Systems), and 20% L-cell conditioned medium. These cells were cultured in petri dishes (low adherence) in a 37°C incubator while on a horizontal rocker at one cycle/3 s for 7 days to avoid attachment and aggregation of newly forming embryoid bodes. On day 8, the floating embryoid bodies were transferred to gelatin coated P-100 tissue culture plates in MDM. Five days later, floating macrophage progenitor cells were harvested and filtered through a 30 μm sterile filters (Sysmex, 04-004-2326) to remove any embryoid bodies, and plated on gelatin-coated tissue culture plates. This harvest of macrophage progenitors was repeated every other day. In order to determine the efficacy of differentiation into macrophages, we performed a DiI labeled acetlylated low-density lipoprotein (DiI-AcLDL) uptake 13 days after plating the macrophage progenitors. Cells were incubated with DiI-AcLDL for 30 min at 37°C and uptake was confirmed by fluorescent-microscopy as previously described (*Hai et al., 2018*). In addition, we compared ESDM with BMDM and found that they were similar by flow cytometry using antibodies against common mouse leukocyte markers: CD11b$^+$, CD11c$^+$, Ly6G$^-$, and Ly6C$^{lo}$. We also determined that undifferentiated DBA/2 ES cells were CD11b$^-$ and CD11c$^-$.

## Bioinformatic analysis

Genes in QTL intervals were determined by custom written R functions ('QTL_gene' and 'QTL_summary') which utilize publicly available BioMart data from Mouse Genome Build 37. A custom written R function ('pubmed_count'), which utilizes the rentrez package in R was used to determine the number of PubMed hits for Boolean searches of gene name and term of interest. Custom written R functions ('sanger_AKRvDBA_missense_genes' and 'missense_for_provean') were used to determine the number of missense (non-synonymous) mutations between AKR/J and DBA/2J mice in QTLs, as documented by the Wellcome Trust Sanger Institute's Query SNP webpage for NCBIm37 (https://www.sanger.ac.uk/sanger/Mouse_SnpViewer/rel-1211). Custom written VBA subroutines ('Provean_IDs' and 'Navigate_to_PROVEAN') were used to automate PROVEAN software (http://provean.jcvi.org/seq_submit.php) queries for predicted functional effects of missense mutations in each QTL, with rentrez functions utilized to retrieve dbSNP and protein sequence data. Ultimately, custom R code was used to generate output tables. Deleterious mutations were designated as defined by PROVEAN parameters (*Choi and Chan, 2015*). All custom written code can be found at http://www.github.com/BrianRitchey/qtl, (*Ritchey, 2021*; copy archived at swh:1:rev:9792fef3dfa7ecdd62857d58ca3f9966456ae6b8).

## Additional information

### Funding

| Funder | Grant reference number | Author |
| --- | --- | --- |
| National Institutes of Health | P01 HL029582 | Jonathan D Smith |
| Lerner Research Institute, Cleveland Clinic | | Jonathan D Smith |

The funders had no role in study design, data collection and interpretation, or the decision to submit the work for publication.

## Author contributions
Brian Ritchey, Conceptualization, Data curation, Software, Formal analysis, Investigation, Methodology, Writing - original draft, Writing - review and editing; Qimin Hai, Conceptualization, Investigation, Methodology, Writing - original draft, Writing - review and editing; Juying Han, Investigation, Writing - review and editing; John Barnard, Software, Formal analysis, Investigation, Methodology, Writing - review and editing; Jonathan D Smith, Conceptualization, Data curation, Formal analysis, Supervision, Funding acquisition, Validation, Investigation, Methodology, Writing - original draft, Project administration, Writing - review and editing

## Author ORCIDs
Jonathan D Smith (iD) https://orcid.org/0000-0002-0415-386X

## Ethics
Animal experimentation: All animal studies were approved by the Cleveland Clinic Institutional Animal Care and Use Committee, protocols 1346 and 1898.

## Decision letter and Author response
Decision letter https://doi.org/10.7554/eLife.68203.sa1
Author response https://doi.org/10.7554/eLife.68203.sa2

# Additional files

## Supplementary files
• Source data 1. Source data in file *Figure 5*. Unedited western blots in *Figure 5D* unedited western blot source data.docx.

• Supplementary file 1. Supplementary file. (a) Genes within the Irm3 QTL interval. All genes in the interval are shown along with the Mb position on chromosome 7. Other columns show distance from the LOD peak in Mb, whether a cis-eQTL was found and if so the LOD score for the eQTL, PubMed hits for the gene and the terns 'inflammasome' and 'IL-1b', the number of non-synonymous SNPs between the AKR and DBA/2 mouse strains, and the PROVEAN analysis to determine the number of non-synonymous SNPs likely to be deleterious to protein function. a, LOD score for cis-eQTL based on our prior BMDM strain intercross (Reference: J Hsu and JD Smith, PMID: 23525445 DOI: 10.1161/JAHA.112.005421); b, total number of PubMed hits for Boolean queries of the respective gene name and terms of interest.; c, non-synonymous SNPs between AKR and DBA/2 mice; d, PROVEAN, number of SNPs predicted to be deleterious by PROVEAN software. Yellow highlighting, top candidate gene. (b) *Pycard* gene sequencing PCR primer pairs. The sequence of the 6 PCR primer pairs used for Sanger sequencing of the *Pycard* gene in AKR and DBA/2 genomic DNA, along with the position of the primers relative to the transcription start site (TSS).

• Transparent reporting form

## Data availability
All data generated for this study are available in Dryad, https://doi.org/10.5061/dryad.wh70rxwn1. All custom written computer code can be found at https://github.com/BrianRitchey/qtl (copy archived at https://archive.softwareheritage.org/swh:1:rev:9792fef3dfa7ecdd62857d58ca3f9966456ae6b8)

The following dataset was generated:

| Author(s) | Year | Dataset title | Dataset URL | Database and Identifier |
|---|---|---|---|---|
| Smith JD | 2021 | Genetic and cellular/biochemical data for mouse Pycard gene differences altering mRNA turnover and IL-1b release | https://doi.org/10.5061/dryad.wh70rxwn1 | Dryad Digital Repository, 10.5061/dryad.wh70rxwn1 |

The following previously published dataset was used:

| Author(s) | Year | Dataset title | Dataset URL | Database and Identifier |
|---|---|---|---|---|
| Hsu J, Smith JD | 2013 | Genetic-genomic replication to identify candidate mouse atherosclerosis modifier genes | https://www.ncbi.nlm.nih.gov/geo/query/acc.cgi?acc=gse35676 | NCBI Gene Expression Omnibus, GSE35676 |

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
