## [Decision Letter]

**Acceptance summary:**

This study highlights that a single nucleotide change in the gene encoding for the universal inflammasome adaptor protein ASC regulates mRNA stability of Pycard and thereby inflammasome function. The authors show that a single SNP in the Pycard gene sequence between AKR and DBA/2 mice causes variation in inflammasome activity.

**Decision letter after peer review:**

Thank you for submitting your article "Genetic Variant in 3' Untranslated Region of the Mouse Pycard Gene Regulates Inflammasome Activity" for consideration by *eLife*. Your article has been reviewed by 2 peer reviewers, and the evaluation has been overseen by a Reviewing Editor and Tadatsugu Taniguchi as the Senior Editor. The following individual involved in review of your submission has agreed to reveal their identity: Eicke Latz (Reviewer #1).

Essential Revisions:

1) Please add quantification of ASC speck formation provided in Figure 2C. This is easily done with image analysis software.

2) ASC is highly regulated on the posttranscriptional and posttranslational levels. For example, in humans and mice, phosphorylation of ASC regulates its activity, and in humans, elaborate splicing of the gene occurs, which regulated inflammasome activity. These points could be discussed to broaden the implications of this study.

*Reviewer #1:*

Genetic differences in outbred species such as humans and differences in the epigenomic structure form the basis of the large variability in the immune response. In particular, the inflammasome is highly regulated at multiple levels, including the post-transcriptional and post-translational levels. Inflammasome responses towards a myriad of triggers are associated with disease development in murine models of disease. Furthermore, clinical trials are ongoing testing the ability of inflammasome inhibitory small molecules to prevent or ameliorate inflammasome-driven pathologies in patient populations.

This manuscript identified that a single nucleotide change in the gene encoding for the universal inflammasome adaptor protein ASC regulates mRNA stability of Pycard and thereby inflammasome function. A particular strength of this manuscript is that the authors managed to show, using genetic alterations, that the single SNP in the Pycard gene sequence (rs33183533) between AKR and DBA/2 mice is the cause of variance in inflammasome activity. Given the relevance of inflammasome for various human pathologies, this work is important for a broad readership.

*Reviewer #2:*

In this manuscript, Ritchey and colleagues studied an intercross of two inbred mouse strains for their inflammasome response to interrogate the genetic basis for enhanced inflammasome activity. This was spurred by the observation that bone marrow-derived macrophages (BMDM) from DBA/2 mice showed an approximately 2-fold enhanced NLRP3 inflammasome response compared to BMDMs from AKR mice. To explore this phenomenon, they stimulated BMDMs from DBA/2 and AKR intercrosses (F4 generation) with NLRP3 agonists and then studied the ensuing IL-1β response. Conducting quantitative trait locus (QTL) mapping the authors then identified a region on chromosome 7 to have the highest LOD score for the phenotype studied (this region was named Irm3). The Irm3 region encompasses the 134.80-138.45 Mb interval on chromosome 7 that encodes for 66 genes. Given its established role in inflammasome signaling and also a strong cis eQTL LOD score, the authors focused on Pycard in the following. Comparing the two mouse strains, the authors noted an SNV in the 3' UTR of the Pycard gene with differing genotypes for DBA/2 and AKR mice. This SNV is located just downstream the stop codon, a region that seems to display little conservation across different mammalian species. Comparing ASC protein expression, the authors noted increased levels of ASC in BMDMs from DBA/2 mice, a finding that also translated into higher amounts of ASC speck levels following inflammasome stimulation. Subsequent experiments indicated that Pycard mRNA levels of BMDMs from DBA/2 mice displayed a longer half-life, while Pycard mRNA transcription or splicing was not affected. Modeling the 3' UTR region of interest furthermore suggested that the SNV impacts on the structure of this region. To validate the causal role of this SNV in regulating Pycard expression, the authors generated DBA/2 ES cells, in which they changed the genotype of this SNV into the corresponding AKR variant. Comparing ES-cell-derived macrophages of the parental DBA/2 genotype to the AKR-adapted Pycard genotype, the authors found that ASC expression levels were indeed decreased and that this reduced expression translated into a reduced NLRP3 inflammasome response in these cells. Altogether, these data suggest that an SNV in the 3' UTR of the murine Pycard gene impacts the stability of its mRNA, which translates into altered ASC protein levels and thereby the activity of inflammasome pathways.

Strength:

The conclusions of this paper are well supported by data and there are no major gaps or flaws in the line of reasoning. A particular stronghold is the functional validation of the here-identified SNV using a CRISPR-based point mutagenesis approach. This set of data provides a high level of confidence for the proposed model.

Weakness:

While this manuscript provides an elegant QTL mapping approach to identify differential expression of Pycard as a major regulator of inflammasome activity in murine BMDMs, the outcome of this study does not provide any new biological insight into inflammasome biology. The fact that differential expression of ASC impacts on inflammasome activity is well expected based on its firmly established role in inflammasome signaling.

Unfortunately, the here-identified mechanism of the differential regulation of the half-life of the Pycard mRNA is not conserved in other species, which precludes any extrapolations to other organisms. Moreover, as also correctly summarized by the authors, there is currently no evidence that genetic variants leading to differential ASC expression in humans would impact on human health or disease. These shortcomings obviously limit the conceptual advance and relevance of the here-identified mechanism.

---

## [Author Response]

Essential Revisions:1) Please add quantification of ASC speck formation provided in Figure 2C. This is easily done with image analysis software.

We added the following sentence to the Legend of Figure 2c: “For the fields shown in the lower panels, primed and activated AKR BMDM yielded 2 specks among 98 nuclei, while DBA/2 BMDM yielded 23 specks among 58 nuclei.”

2) ASC is highly regulated on the posttranscriptional and posttranslational levels. For example, in humans and mice, phosphorylation of ASC regulates its activity, and in humans, elaborate splicing of the gene occurs, which regulated inflammasome activity. These points could be discussed to broaden the implications of this study.

We added the following paragraph to the Discussion section, and made a new joiner to the final paragraph: “ASC protein/PYCARD gene have been previously shown to be subject to additional regulatory control by post-translational modifications such as ubiquitination and phosphorylation and by mRNA splicing (21). The tyrosine kinase Syk is required for Nlrp3 inflammasome activation in mouse BMDM (22). Both Syk and *Jnk* kinases are required for Nlrp3 and AIM2 inflammasome activation in mouse and transfected human cells, which are associated with ASC phosphorylation and oligomerization (23). Human ASC tyrosines 146 and 187 are critical for Syc mediated ASC oligomerization and inflammasome activity (24). Syk phosphorylates Pyk2 kinase, which then directly phosphorylates ASC on tyrosine 146 (25). In contrast IkB kinase a (IKKa) binds directly to ASC, requiring ASC serines 16 and 193, and negatively regulates Nlrp3 inflammasome activity, so that upon inflammasome activation by ATP or other agonists IKKa is released from ASC (26). Protein phosphatases also regulate ASC oligomerization and activity, with the broad spectrum tyrosine phosphatase inhibitor phenylarsine oxide reducing ASC assembly and speck formation (27); although this appears to be in opposite direction from the Syk effects on ASC. The protein tyrosine phosphatase PTPN2 is a negative regulator of inflammasome activity, as demonstrated in myeloid specific Ptpn2 knockout mice that have increased inflammasome activity and ASC phosphorylation via increased *Jnk* activity (28). Based on cDNA sequences, RT-PCR, and ASC domain specific antibodies, alternative splicing of PYCARD mRNA has been proposed leading to 3 alternative human ASC isoforms; and, transfection overexpression studies found some inflammasome activity with the alternative exon 2 skipped isoform (29).

Our study identified an additional ASC regulatory mechanism in mice due to a *Pycard* 3’ UTR SNP, where the allele that decreases *Pycard* mRNA stability was found in ~1/3 of sequenced mouse strains.”